# Using Mouse and *Drosophila* Models to Investigate the Mechanistic Links between Diet, Obesity, Type II Diabetes, and Cancer

**DOI:** 10.3390/ijms19124110

**Published:** 2018-12-18

**Authors:** Coral G. Warr, Katherine H. Shaw, Arani Azim, Matthew D. W. Piper, Linda M. Parsons

**Affiliations:** 1School of Biological Sciences, Monash University, Clayton, Victoria 3800, Australia; coral.warr@utas.edu.au (C.G.W.); katherine.shaw@monash.edu (K.H.S.); arani.azim@monash.edu (A.A.); matthew.piper@monash.edu (M.D.W.P.); 2School of Medicine, College of Health and Medicine, University of Tasmania, Hobart, Tasmania 7000, Australia

**Keywords:** cancer, diet, obesity, type II diabetes mellitus, *Drosophila*

## Abstract

Many of the links between diet and cancer are controversial and over simplified. To date, human epidemiological studies consistently reveal that patients who suffer diet-related obesity and/or type II diabetes have an increased risk of cancer, suffer more aggressive cancers, and respond poorly to current therapies. However, the underlying molecular mechanisms that increase cancer risk and decrease the response to cancer therapies in these patients remain largely unknown. Here, we review studies in mouse cancer models in which either dietary or genetic manipulation has been used to model obesity and/or type II diabetes. These studies demonstrate an emerging role for the conserved insulin and insulin-like growth factor signaling pathways as links between diet and cancer progression. However, these models are time consuming to develop and expensive to maintain. As the world faces an epidemic of obesity and type II diabetes we argue that the development of novel animal models is urgently required. We make the case for *Drosophila* as providing an unparalleled opportunity to combine dietary manipulation with models of human metabolic disease and cancer. Thus, combining diet and cancer models in *Drosophila* can rapidly and significantly advance our understanding of the conserved molecular mechanisms that link diet and diet-related metabolic disorders to poor cancer patient prognosis.

## 1. Introduction

The effect of nutrition on cancer has received much attention, spawning an industry of books and websites purporting a vast array of controversial dietary information and supplements for cancer patients. A plethora of ‘recommended’ diets exist, ranging from those that claim to prevent cancer to those suggesting food types and supplements that enhance the effectiveness of therapy and prevent relapse [1,2]. Often dietary guidelines are contradictory and subject to change, promoting anxiety in the general public, cancer patients and their carers [3]. Overall, although identification of the precise dietary components that influence cancer biology is controversial, one clear finding has emerged: individuals with diet-related metabolic disorders, such as obesity and type II diabetes, have an increased cancer risk. In addition their cancers are more aggressive and more resistant to current therapies, compounding the prognosis of these patients [4,5,6,7,8]. 

The earliest suggestion of a link between diet and cancer arose from laboratory mouse studies conducted in the 1940s [9]. These early reports demonstrated that underfeeding, by restricting dietary intake of calories by approximately one third, caused a significant reduction in tumor incidence and tumor growth in mice. This led to the suggestion that perhaps all cancers respond to dietary influence. The first compelling human epidemiological studies to reveal a causal relationship between diet and cancer noted changes in cancer incidence in migrant populations. Notably, Japanese women have a ten-times lower incidence of breast cancer than do residents of Western countries, but when these women migrated to Hawaii, their incidence of breast cancer rose to a similar level as found in the Caucasian Hawaiian population [10]. Later studies highlighted variability in rates of specific cancers between different countries and gave rise to the concept of the healthy ‘Mediterranean diet’ [11] amongst others. 

There have since been numerous epidemiological studies assessing the effect of various specific dietary factors on the incidence of cancer [12]. Taken together, these suggest that alcohol, fat, and red meats are associated with increased risk of colorectal, liver, and breast cancers, whilst a range of fruit and vegetables including red grapes, garlic, and turmeric offer protective effects [12,13,14,15]. Many of the early studies focused on a single dietary component, such as resveratrol, which is found in red grapes, peanuts, and berries. These studies revealed conflicting and/or inconclusive findings [15]. 

Nevertheless, in the last decade one clear theme has emerged: obesity and obesity-related diseases, in particular type II diabetes, are associated with increased risk of colon, uterine, liver, pancreatic, breast (in post-menopausal women), and prostate cancers amongst others [16]. In 2012, estimates claimed that obesity contributed to an additional 3.6% (or 481,000) of all new cancer cases worldwide (excluding melanoma), with the greatest burden on Europe, North America, and Australia [17]. In breast cancer, excess weight and obesity also drive cancer progression, and are associated with recurrence [18,19], metastasis to bone and poor response to chemotherapeutic agents [20]. Whilst an association between obesity/type II diabetes and sub-types of prostate cancer is unresolved [21,22], there is strong evidence for a relationship between obesity and an elevated risk of aggressive prostate cancer [23]. Cancers of the breast and prostate are some of the most commonly diagnosed, and are the leading causes of cancer death in women and men, respectively. Moreover, the largest cohorts of cancer survivors are those treated for breast and prostate cancer, yet those who additionally suffer obesity have increased chance of relapse and respond poorly to further cancer therapies [19]. 

Despite the potential impact of diet on cancer, and its associated economic burden, the mechanistic links between diet, dietary related metabolic disorders, and cancer remain poorly understood. Such mechanistic studies cannot be conducted using in vitro tissue-engineered tumor models because these fail to mimic the interactions between tumor cells, organ systems and diet that occur in whole animals. Thus, a significant barrier to current cancer research is the need to use whole animal models. Transgenic mouse models that manipulate nutrient status and cancer are in their infancy [24] and complex mouse models genetically targeting metabolic and cancer pathways are both technically challenging to develop and expensive to maintain [25]. The ability to readily manipulate the *Drosophila melanogaster* genome in order to alter gene function, either throughout the whole animal or within specific cell types, allows the molecular mechanisms of gene function to be characterized at a level of resolution not possible in current vertebrate models. Additionally, the low cost and rapid generation of multiple genetic and cell biology tools in *Drosophila* results in efficient in vivo investigation of the metabolic and growth pathways that underlie tumorigenesis. The conservation of fundamental biology and physiology between flies and humans, has allowed *Drosophila* researchers to deliver pivotal breakthroughs in our understanding of tumor biology [26,27,28]. Therefore, innovative models using a simpler animal system, the vinegar fly *Drosophila melanogaster*, may increase our capacity to make a timely and relevant contribution to understanding the relationships between cancer, diet, and dietary-induced metabolic disorders.

## 2. Modeling Interactions between Diet and Cancer in Mice

### 2.1. Modeling Cancer in Mice

To gain knowledge of the links between diet and cancer, animal models in which the aetiology of human cancers can be recapitulated whilst altering dietary inputs and monitoring metabolic and tumor biology are required. The laboratory mouse has a long history of providing fundamental insights into the mechanisms of cancer initiation, progression and metastasis [29]. Two major tools for cancer studies in mice are xenografts and genetically-engineered mouse models (GEMM), both of which have advantages and disadvantages for studying human cancers. 

The current gold standard of xenograft models is the ‘human-in-mouse’ xenografts (i.e., patient-derived xenografts) in which advanced tumor fragments or metastases are implanted into an immunodeficient mouse subcutaneously, or, with greater difficulty, orthotopically (i.e., into the organ of interest). The major advantage of this system is that initially tumors retain human-specific microenvironment features, which are more reflective of the histology, natural architecture and genetic heterogeneity seen in primary tumors [30]. Unfortunately, a major obstacle to patient-derived xenograft modeling is the extremely poor engraftment rate of common tumor types, such as estrogen receptor-positive breast and prostate cancers [31]. In addition, and importantly, patient-derived xenografts may not completely recapitulate tumor heterogeneity as the cells that grow may only represent a subpopulation of tumor cells. Another limitation of xenograft cancer models is that tumor growth occurs in a host with an impaired immune system, thereby bypassing the anti- and pro-tumor activity of the adaptive immune system. Thus, although the patient-derived xenograft cancer models are essential tools for in vivo assessment of cancer biology, their utility for modeling diet and cancer interaction is limited. 

Genetically-engineered mouse models circumvent several issues that arise using patient-derived xenograft models because tumors arise *in situ* where inflammatory processes can interact with the developing tumor. Ideal GEMMs of human cancers feature tumors that carry the corresponding human mutation and that arise in a small subset of cells within normal tissue, where the steps in cancer progression (e.g., acquisition of secondary mutations, changes in tumor cell metabolism and ability to metastasize) closely resemble human pathologies [32]. However, a major limitation of GEMMs is that development and validation of these models is time-consuming, laborious, and expensive. This is exemplified when a novel mutation is introduced in an existing multi-allelic mouse model, as this requires extensive breeding. Several strategies have been devised to overcome some of these problems. For example, multiple transgenic constructs, knockouts or knockins can be introduced into embryonic stem (ES) cells that are then used to generate experimental animals [33]. An advantage of this approach is that modified ES cells carrying multiple genetic manipulations can be frozen and stored until needed, greatly reducing the costs associated with maintaining complex mouse strains. Inhalation of viral vectors bearing one or more cDNAs or shRNAs directly in vivo into the adult mouse lung has also proven to be an effective strategy to test the role of multiple genetic changes in tumorigenesis [34]. However, it remains to be determined if viral delivery of genetic material into other adult mouse organs will be as fruitful.

### 2.2. Investigating Interactions between Diet and Cancer in Mice

Despite the substantial human epidemiological evidence that reveals a clear association between cancer and diet, this area of human health is noticeably understudied. Yet, even with such limited study, increased tumor growth, and more aggressive metastatic disease has been observed in animals reared on high calorie diets for a variety of mouse cancer models, including pancreatic [35], prostate [36,37], ovarian [38], and breast cancers [39].

The most well-studied models combining cancer and diet manipulation are colorectal cancer models using xenografts or GEMMs [24,40]. For example, human colon cancer HT-29 cells xenografted under the skin of mice generate tumors that metastasize, accompanied by changes in cell shape and migratory behavior (epithelial to mesenchymal transition (EMT)). When animals bearing HT-29 xenografts were fed a high fat diet this resulted in tumor growth associated with increased activation of the mitogen-activated protein kinase (MAPK) and Phosphatidylinositol 3-kinase (PI3K) pathways, and tumor cells showed accelerated EMT progression [41]. Additionally, as discussed above, this xenograft model is ectopic and was performed in immunodeficient mice. This means that the impact of the known chronic state of site-specific low-grade inflammation that occurs in humans due to the consumption of a high fat diet, which is hypothesised to promote tumor development and growth [42], cannot be examined. Thus, it can be more revealing to combine dietary manipulation with GEMMs of cancer to truly understand the complex interactions between diet and cancer biology. In this regard, the discovery of a mutant mouse with multiple intestinal neoplasia (*Min*) has led to the development of several excellent mouse models to study the early stages of human colon cancer and the effects of dietary manipulation on cancer progression. *Min* mice have a mutation that results in truncation of the adenomatous polyposis coli (Apc) protein, similar to that found in patients with familial adenomatous polyposis and in 80% of patients with sporadic colon cancers [43]. Heterozygous *Apc^Min/+^* mice fed an ‘obesogenic American-diet’ showed increased polyp size, but no change in total polyp number [44]. This report also demonstrated that the ‘American-diet’ altered immune cell behavior in adipose tissue and the tumor microenvironment [44], providing an excellent model system to further explore the systemic impacts of obesity upon tumorigenesis. Conversely, *Apc^Min/+^* mice have also been used to demonstrate that introducing a source of high dietary fiber (rice bran) decreased tumor burden [45]. These studies also identified changes in metabolic pathways and a weak correlation between increased levels of the hormone adiponectin (involved in regulating glucose levels and fatty acid breakdown), and decreased polyp number. Thus, providing new insight into pathways that maybe beneficial for cancer diagnosis or treatment. However, the cellular and molecular relationship between adiponectin levels and tumor biology is still unknown. Thus, establishment of mouse cancer models that display altered cancer progression in response to dietary inputs now provides researchers with the ability to identify the molecular interactions between diet and cancer which impact tumor growth.

It is clear that mouse models can provide invaluable insights into the relationship between diet and cancer progression. However one problem with these studies is the lack of detail provided to describe the diets used in different laboratories. Currently, there are a number of high fat or high sugar diets commercially available such as the ‘Western’, ‘Sweet Stuff’, or ‘Strictly Vegan’ diets from the Jackson Laboratories (Bar Harbor, CA, USA), or the high and low fat diets from BioServ Inc. (Frenchtown, NJ, USA) and Research Diets (New Brunswick, NJ, USA). Furthermore, several research groups use ‘in house’ designer diets such as the ‘American-diet’ [44]. Careful documentation of dietary regimes including specific amounts of macronutrients (such as protein and carbohydrate) and micronutrients (such as vitamins and minerals) together with feeding regimes, is needed to allow meta analyses to assess the roles of different nutrients on tumor progression. These analyses will also shed light on whether calories derived from fats, sugars or proteins impact tumor growth equally. Furthermore, incorporating the genetic background of mice into meta-analyses will enable the interactions between cancer, diet, and genes to be examined. Finally, for mouse cancer models to be of maximum benefit to human health, mouse models designed to capture the complexities of diet-related human diseases coupled with dietary intervention trials must also be developed. 

## 3. Investigating Interactions between Cancer, Type II Diabetes Mellitus, and Obesity in Mice

Although the mechanisms that underlie increased susceptibility to cancer in individuals with type II diabetes or obesity remain largely unknown, several concepts are beginning to emerge through the study of mouse models that combine cancer with a model of obesity/ type II diabetes (generated by a genetically engineered deletion of the Insulin Growth factor 1 (IGF1)) [46,47]. For example, tumor progression from xenografts taken from a variety of mouse transgenic breast cancer strains (mouse mammary tumor virus (MMTV) -Neu, c-Myc or vascular endothelial growth factor (VEGF)) is accelerated in IGF1 deficient mice [46,48]. Cancers in these animals displayed a number of known essential mechanisms for tumor cell proliferation, survival, and metastasis. These include sufficiently elevated insulin levels to promote InR signaling, increased levels of the powerful oncogene c-Myc, increased angiogenesis via VEGF, and elevated levels of the extra cellular matrix degradation protein matrix metalloprotease-9 [49,50]. Further studies of cancer progression in mice deficient for IGF1 suggest that these mechanisms of tumor growth may be applicable to colon [51] and pancreatic [52,53] cancers. Taken together, mice with a genetically engineered IGF1 deletion have increased the capacity of researchers not only to understand how type II diabetes contributes to tumor growth, but also how this complex multifactorial disease impacts the tumor microenvironment and promotes metastasis. 

Recent studies have begun to shed light on the potential molecular mechanisms that may link obesity with aggressive cancer progression and poor outcomes in breast cancer patients [54]. Emerging evidence suggests elevated levels of circulating adipose fatty acid binding protein (A-FABP) in obese patients correlates with elevated alcohol dehydrogenase 1 (ALDH1) levels in breast tumor cells and also with poor prognosis in obese breast cancer patients [54,55,56]. Injection of in vitro cultured breast tumor cells under the mammary fat pad of wildtype mice resulted in the release of A-FABP from adipose tissues and a concomitant increase in the expression of ALDH1 in transplanted tumor cells. Importantly, in GEMM that lacked *A-FABP*, ALDH1 expression levels and tumor growth were reduced in transplanted breast tumor cells. Furthermore, *A-FABP^-/-^* mutant mice bearing orthotopically injected breast cancer cells fed a high fat diet show reduced tumor burden compared to *A-FABP^-/-^* mice on a low fat diet. Finally, in a GEMM of post-menopausal breast cancer (MMTV-Transforming Growth Factor) loss of *A-FABP* also showed reduced ALDH1 positive cells and decreased tumor growth [54]. These results suggest that circulating A-FABP enhances the aggressiveness of mammary tumor cells in both transplantation and transgenic mouse models. A-FABP activates the Interleukin 6/Signal Transducer and Activator of Transcription 3/ALDH1 pathway within breast tumor cells which promotes a stem cell phenotype that enhances the aggressiveness of these tumor cells [54]. Thus, A-FABP may represent a new link between obesity and increased risk of breast cancer and provide new diagnostic or therapeutic opportunities for patients.

## 4. Using *Drosophila* to Model Interactions between Diet, Diet-Related Disease, and Cancer 

Despite progress using mice to model diet, obesity-type II diabetes, and cancer interactions, significant issues remain with these models. Predominantly, mouse models that manipulate nutrient status are in their infancy, and complex mouse models targeting multiple cancer pathways are technically challenging and expensive to maintain. We argue that the vinegar fly *Drosophila melanogaster* is an excellent model for addressing fundamental aspects of the interaction between diet and cancer. They are inexpensive to maintain, small, and have a simple diet, and therefore a vast number can be kept in the laboratory. Their short lifecycle (9–10 days) and exemption from many regulatory considerations ensures rapid and timely progress can be made, particularly with respect to cancer studies that can take several months or years to complete in mice. In addition, *Drosophila* researchers have developed powerful genetic techniques and dietary models that allow for the rapid identification and characterization of genes involved in multiple human diseases, which include cancer and metabolic disorders [57,58]. In particular, *Drosophila* offers unparalleled opportunities for genetically manipulating multiple genes and pathways in a tissue-specific manner (needed for cancer studies) within a whole animal model (needed for dietary studies). Second, the similarity of cellular processes such as growth and metabolism and the emerging evidence of functional conservation of genes between *Drosophila* and mammals mean that studies in flies can directly contribute to the understanding of human disease [59,60]. 

Notwithstanding the advantages of using *Drosophila* to investigate human cancers and metabolic disorders, there are also several limitations that should be considered. First, a clear physiological difference between flies and mammals is the optimal body temperature. Flies are ectotherms and can be successfully reared between 18 and 27 °C whilst humans are endothermic and closely maintain a body temperature of approximately 37 °C. As ectoderms do not have to maintain body temperature it has been proposed that these animals expend more energy on growth and reproduction than endoderms [61]. Hence, the relationship between diet, energy expenditure, and animal growth between flies and humans may be biased towards different physiological outcomes. Second, flies lack a clear equivalent of breast, prostrate, and lung tissues, thus preventing directly comparable models of these cancers from being generated. Nevertheless, human epithelial and stem cell cancers have been successfully modeled in *Drosophila* [26]. *Drosophila* also lack a closed circulatory system rendering the modeling of tumor induced angiogenesis and associated changes in the tumor microenvironment difficult. Finally, *Drosophila* models of drug discovery may be limited due to differences in drug metabolism pathways between flies and humans [60,62].

### 4.1. Modeling Diet-Related Human Metabolic Diseases in Drosophila

Although humans and flies differ greatly in terms of their gross morphology, many of the digestive and endocrine systems that control nutrient uptake, storage, and metabolism in humans are also present in *Drosophila* [63,64]. Food is digested and absorbed in the fly proventriculus and midgut as in the mammalian stomach and intestine [65]. Key organs that regulate major metabolic pathways and energy storage, such as the liver and pancreas in mammals, are also conserved in the larva and adult fly as oenocytes, fat bodies, and neuronal insulin producing cells (iPCs) respectively [66,67] (Figure 1). 

In mammals, glucagon and insulin secreted by pancreatic α and β cells respectively, maintain blood sugar levels. In *Drosophila*, endocrine cells located anterior to the brain secrete Adipokinetic hormone (AKH) that is equivalent to glucagon, whilst iPCs located within the fly brain are analogous to pancreatic cells (Figure 1).

In flies, the fat body takes on the dual role of liver and adipose tissue, storing energy in the form of both glycogen and lipids and releasing energy in the form of the glucose disaccharide trehaolse. Similar to the liver, the fat body also serves as an endocrine organ, secreting peptides such as insulin-like peptide 6 (iLP6) to coordinate metabolic homeostasis [68]. One target of secreted iLP6 is the oenocytes which have a critical role in fat mobilization and turnover in starved larvae. Oenocytes accumulate lipid droplets during starvation to release ketones and express many genes with homology to liver-specific enzymes in mammals. These traits have led to the concept that oenocytes are equivalent to liver hepatocytes [23,69,70]. 

The biochemical pathways responsible for regulating appropriate circulating sugar levels and energy storage in mammals are also highly conserved in *Drosophila* [71,72]. Conserved regulators of nutrient transport [73], cellular sugar and amino acid flux [74], mitochondrial energy pathways [75], the hormones insulin and glucagon, and signaling pathways such as Insulin Receptor-mammalian Target Of Rapamycin (InR-mTOR) share sequence, structural, and functional similarities with vertebrate homologues [76]. In mammals, IGFs regulate growth via Insulin Growth Factor Receptors, whereas insulin maintains glucose homeostasis via the InR. However, in flies the single InR regulates both metabolism and animal growth [77]. Thus, *Drosophila* models can be used to investigate various aspects of vertebrate metabolic function including diet, nutrient uptake and the regulation of energy storage or expenditure.

Importantly, *Drosophila* larvae reared on a high sugar diet (HSD, 34% sucrose) or a high fat diet (HFD, 20% coconut oil) exhibit hallmarks of human type II diabetes [78,79]. These animals display high circulating trehalose levels (hyperglycemia), with increased insulin like peptide 2 (*iLP2*) mRNA, and protein expression, and elevated circulating iLP2 levels. Importantly, larvae raised on a HSD or HFD supplemented with insulin show reduced responses to insulin and diminished activation of the InR pathway target Protein Kinase B (Akt), indicating *Drosophila* larvae raised on a HSD or HFD display insulin resistance, which is a defining feature of type II diabetes. Finally, insulin resistant diabetes is often present in obese patients, and larvae raised on a HSD and HFD accumulate stored fat in the form of triacylglycerides and free fatty acids [78,79,80]. Thus, it appears that the links between diet, obesity and type II diabetes are evolutionarily conserved, suggesting that *Drosophila* is an excellent model to explore the genetic and molecular mechanisms that link dietary fats and sugars to human diseases [81].

### 4.2. Investigating Interactions between Diet and Cancer in Drosophila

Since the identification of the first tumor suppressor mutations in *Drosophila* [82], flies have provided pivotal breakthroughs in our understanding of cancer biology. These seminal contributions are highlighted in recent reviews [26,28], therefore we focus here on how *Drosophila* studies are shedding light on the relationship between cancer and diet. In flies, it is possible to generate groups of tumor cells in patches of epithelial tissue surrounded by a normal cellular and signaling environment, thus mimicking the sporadic nature of human tumorigenesis [83]. Furthermore, hemocytes (invertebrate innate immune cells) are recruited to tumors, modeling anti- and pro- tumor interactions with the adaptive immune system [84]. In one study that investigated the impact of diet, the proliferation of cells bearing a loss of function mutation in the tumor suppressor gene Phosphatase and tensin homolog (*PTEN*, a negative regulator of PI3K signaling) was shown to be increased under conditions of nutrient restriction (e.g., caloric restriction of protein by 45% and sugar by 25%) [85]. Thus, *PTEN* mutant cells increase tumor growth via cell multiplication under conditions of amino acid starvation and energy stress. The increased proliferation of *PTEN* mutant cells facing severe nutrient restriction comes at the expense of neighboring wildtype cells. *PTEN* mutant tissue is metabolically more active and outcompetes the surrounding wildtype tissue for nutrients via the InR pathway. In addition, under conditions of nutrient restriction *PTEN* mutant cells also sustain their energy needs by systemically reducing the growth of organs throughout the entire animal [85]. The mechanisms behind the reduction in organ growth are as yet unknown. These studies clearly demonstrate that diet-cancer cell interactions can influence the survival of neighboring normal tissue and organ growth throughout the entire animal.

### 4.3. Investigating Interactions between Dietary Related Metabolic Disorders and Cancer in Drosophila

The laboratory of Ross Cagan was the first to establish a *Drosophila* sugar-enhanced cancer model (SECM) [86]. This model capitalizes on the powerful molecular genetic approaches possible in *Drosophila* to model human tumor development, and the ability of a high sugar diet (HSD) to induce type II diabetes in growing larvae. *Drosophila* SECM tumor cells contain the *csk^Q156STOP^* mutation that results in activation of the potent oncogene Src. In addition, these tumor cells also overexpress an activated isoform of the small GTPase *Ras* (*Ras^V12^*). This mimics the elevated Src and Ras signaling observed in a number of human cancers including breast, colorectal and pancreatic cancers [87]. In larvae raised on a low sugar diet (10% sucrose), *Ras^V12^*/Src co-activated cells develop small, slow-growing tumors within the epithelial tissue. By contrast, in larvae raised on a HSD the *Ras^V12^*/Src co-activated cells develop large primary tumors, as well as metastatic secondary tumors due to increased Wingless (Wnt in mammals) signaling [86]. Thus, raising tumor-bearing larvae on a HSD leads to increased tumor growth and metastatic spread. 

A major advantage of the *Drosophila* SECM is the ease with which secondary mutations that alter tumor growth can be then introduced into these flies, enabling mechanistic studies of links between diet and cancer signaling pathways to be performed. For example, the Cagan lab showed that members of the conserved AMP activated kinase family, *Drosophila* Salt Inducible Kinases 2 and 3 (SIK2 and 3), were required for tumor growth specifically on a high sugar diet [88]. These studies showed that SIKs regulated tumor growth via the conserved Hippo signaling pathway. However, the role of SIKs in tumor cell metabolism was not examined, which is important to do because in both flies and mammals SIKs are well characterized as critical regulators of cell metabolism, specifically lipid and glucose homeostasis [89,90,91,92,93]. 

Studies in *Drosophila* show that SIK3 improves animal survival when exposed to increased dietary sugar levels [94]. SIK3 directly phosphorylates and activates glucose-6-phosphate dehydrogenase to promote catabolism of glucose via the pentose phosphate pathway (PPP) [94]. The PPP is an important source of reducing power in the form of NADPH, which is thought to alleviate sugar-induced oxidative stress by maintaining capacity to reduce the cellular antioxidant glutathione [94]. In contrast to normal cells, which primarily rely upon mitochondrial oxidative phosphorylation to generate energy, most tumor cells rely on anaerobic glucose catabolism (the Warburg effect) [95,96]. We do not know what the metabolic role of SIKs in tumor cells undergoing anaerobic glucose catabolism is, nor do we know if high sugar levels impact the activity of the SIKs in tumor cells. This is of interest because human SIKs (hSIKs1-3) were recently discovered to be oncogenes and are attractive candidates for the treatment and diagnosis of cancer [97]. 

In ovarian cancers, 85% of tumor samples show elevated hSIK3 levels [98]. In addition, overexpression of hSIK2 or hSIK3 in human ovarian and breast cell lines demonstrated that hSIKs promote cell proliferation, whilst knockdown of hSIK2 or hSIK3 revealed these kinases are required for tumor cell proliferation [98,99,100]. Further studies demonstrated that, within cells of the tumor microenvironment, increased hSIK2 or hSIK3 activity induces changes in fatty acid and arginine metabolism, which promoted metastasis and growth of secondary tumors [100,101]. These findings highlight that SIKs can orchestrate both metabolic and growth pathways in tumor cells to promote cell survival, proliferation and metastasis. Thus, the conserved SIKs provide a unique and exciting entry point for the dissection of the molecular mechanisms that coordinate diet, cell metabolism and tissue growth. Studying SIKs in *Drosophila* and mouse cancer models that also recapitulate diet, obesity and/or type II diabetes interactions will be crucial to identify growth signaling and/or metabolic pathways that may provide new avenues for cancer diagnostics and therapies.

### 4.4. Development of Dietary Regimes in Drosophila to Study Interactions between Diet and Cancer

Much of the cancer-related dietary research in both *Drosophila* and mice has traditionally focused on caloric restriction, where energy intake is reduced by decreasing the amount of food (diluting food) to around 10–50% or increased by over-nutrition. However, diet is complex and there is debate as to whether the effects of caloric restriction and over-nutrition are due to altered calories or altered intake of specific macronutrient proportions; primarily protein, fat and sugar [102,103]. There is growing evidence from studies on a wide range of species that, rather than sugar, fat or protein acting alone, it is the balance between macronutrients that is more important for health. In particular, in flies and mice it has emerged that the balance of protein to carbohydrate in the diet is especially significant, influencing total energy intake, growth and development, reproduction, and aging [104,105,106,107,108]. The past decade has seen the development of a variety of new dietary tools, including nutritional geometry and the defined diet, to assess *Drosophila* development and growth [109,110,111,112]. Nutritional geometry is of particular interest, as it implements a systematic approach to reveal the effects of macronutrient (e.g., carbohydrate, protein, or fat) interactions with a phenotype of interest. Thus, nutritional geometry is a useful tool to dissect the relative contributions of genes and diet on cancer development.

Nutritional geometry analyses the integrated effects of nutrients and their relative concentrations on biological outcomes, rather than considering particular nutrients in isolation. Individual animals are displayed as separate points in multi-dimensional nutrient space, where the dimensions of that space are defined by particular nutrients of interest (e.g., protein and carbohydrate). The coordinates of the point representing an individual are defined by its nutritional history. Any quantitative phenotype of interest, e.g., animal growth, tumor growth or number of metastases, can be plotted as a surface in the *z*-axis (Figure 2). Statistical models can then be used to assess how phenotypes are affected by individual or interacting effects of the major macronutrients, such as dietary protein and carbohydrate [111]. These studies have provided key insights into the role of macronutrient balance in altering animal physiology. To date no studies have been undertaken in *Drosophila* to monitor tumor growth or metastasis in flies raised on diets with systematically varying protein and carbohydrate ratios. Importantly, such studies are now possible in *Drosophila* given the recent development of defined synthetic diets that enable complete freedom to manipulate the relative proportion of any nutrient in the diet [110]. By combining *Drosophila* cancer models with nutritional geometry studies it will be possible to generate unique experimental paradigms to further define and interrogate how diet influences cancer–diet interactions. 

### 4.5. New Genetic Tools in Drosophila will Further Facilitate Studying Interactions between Diet-Related Metabolic Disorders and Cancer

The workhorse of *Drosophila* genetics is the *UAS*-*GAL4* system, which is used to manipulate gene expression in a tissue and timing-specific manner. This system is based upon the yeast GAL4 transcriptional activator and its DNA target, Upstream Activating Sequence (*UAS*) [113]. A wide variety of *GAL4* driver lines with defined patterns of expression have been generated by many researchers. Furthermore, a large number of *UAS* lines that express *Drosophila* or human genes, or induce RNA interference (RNAi) and therefore reduce gene expression, have been described. These tools underpin multiple experimental approaches that have enabled *Drosophila* researchers to stay at the forefront of biological research [114]. Alternative bipartite Q or *LexA*/LexA_op_ transgene systems have also been developed in *Drosophila* [115,116,117,118]. Combining more than one of these systems will allow the generation of animals in which a variety of genetic manipulations occur independently in multiple tissue types. Thus, we now have the ability to manipulate gene expression separately in distinct organs such as the *Drosophila* equivalent of the liver (fat body) or gut. This builds capacity to undertake systemic approaches in *Drosophila* cancer models, allowing researchers to shed light on the interplay between tumor growth, metastasis, and organ systems throughout the body. By taking advantage of these dual transgene expression systems in *Drosophila* it will now be possible to generate tumors within the fly and manipulate the expression of circulating hormones, and metabolic genes and pathways in specific organs, thereby modeling the complex interactions between cancer progression and metabolic processes in distant organs. *Drosophila* researchers are thus poised to generate new insights into communication between tumor cells and peripheral organs, which is critical for distributing energy stores and maintaining cancer growth. 

## 5. Conclusions

We are currently faced with an epidemic of patients with obesity and type II diabetes. Crucially, these individuals face an increased risk of cancer and decreased survival rates. Resolving the effects of dietary macronutrients on cancer risk and progression remains a fundamental challenge, with profound implications for human health. Animal models that allow precise manipulation of dietary input (nutritional geometry), combined with the capacity to assess nutrient levels, genetically manipulate metabolic and/or growth-signaling networks in specific tissues and/or tumor cells are needed. *Drosophila* provides an excellent model system, as the conservation of fundamental physiology, and metabolic and growth pathways between flies and humans means that studies in *Drosophila* can deliver pivotal breakthroughs in our understanding of nutrition and cancer growth pathways that are applicable to human health.

## Figures and Tables

**Figure 1 ijms-19-04110-f001:**
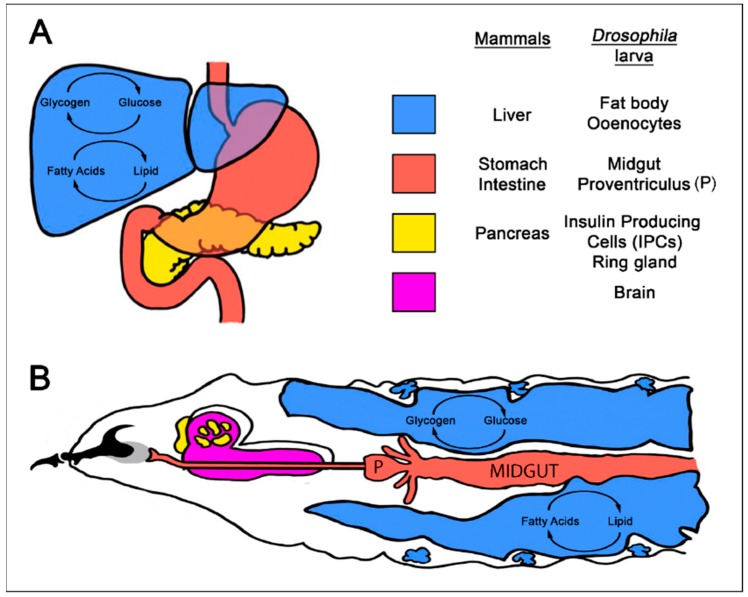
Schematic representation of conserved human and *Drosophila* metabolic organs. Metabolism in *Drosophila* is coordinated by a network of organs that perform the same basic cellular and physiological functions as in humans. (**A**) Frontal view of the major metabolic organs liver, gut and pancreas in humans. (**B**) Dorsolateral view of a *Drosophila* larva with conserved metabolic tissues highlighted. The blue fat body and oenocyte cells perform similar functions to the human liver, whilst the ring gland and insulin producing cells within the larval brain secrete glucagon, insulin and insulin-like peptides respectively to maintain glucose homeostasis.

**Figure 2 ijms-19-04110-f002:**
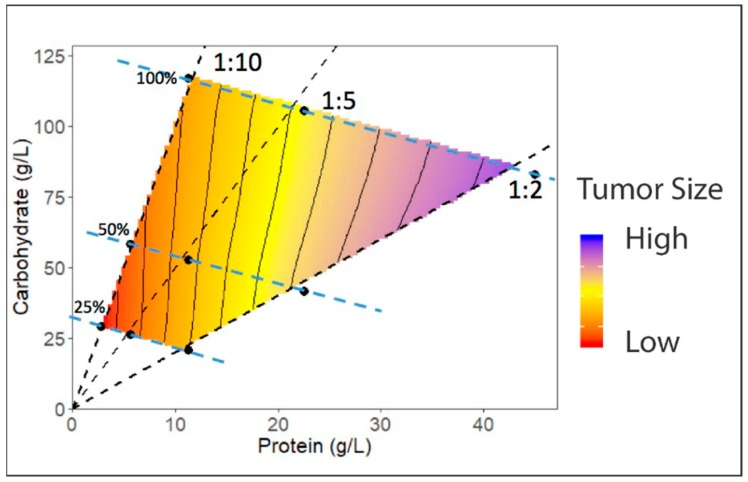
Graphical representation of Nutritional Geometry. A hypothetical interaction between protein (P) and carbohydrate (C) content within the larval diet on tumor size. Larvae fed a diet of both high protein and sugars are predicted to display the greatest levels of tumor growth. The response surface (represented by the colour gradient and thin black contours), generated using thin plate splines, is generated by fitting to tumor size across the nine different diets (filled black circles). These diets consist of one of three P:C ratios (dashed black lines; P:C = 1:2, 1:5, or 1:10) at one of three caloric densities (25%, 50%, and 100%; represented by blue dashed lines). In this case, tumor size is maximised by diets with a high P:C ratio and low caloric density. Tumor size increases as dietary protein increases, with no effect of dietary carbohydrates.

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
