# Peer review of "Using Mouse and *Drosophila* Models to Investigate the Mechanistic Links between Diet, Obesity, Type II Diabetes, and Cancer"

_ijms, 2018, doi:10.3390/ijms19124110_

Round 1

Reviewer 1 Report

The manuscript by Coral and colleagues review animal models that are used in the research on dietary related cancer, obesity and type II diabetes. In the first two parts, the authors described the conventional mouse models used in the current research. By pointing the defect on those models, however, the authors start to focus on the model links between diet and dietary related disease using Drosophila due to their easy on maintenance and study. The manuscript is well written and the authors set an extraordinary model on dietary related disease research by using Drosophila instead of mouse model.
Specific issues: 
1. In the current manuscript, the authors described that obesity is known to increase the risk of developing various types of cancer, especially in breast cancer. A new research published on Cell Metabolism recently indicates that high levels of circulating adipose fatty acid binding protein are associated with high risk of breast cancer. It is well known that FABP family members can affect metabolism, lipid fluxes and signaling in mammalian cells. It must be very valuable to understand which proteins like FABP serve as the same function in Drosophila.

2. As the authors’ claim that the Drosophila model are inexpensive, simple to maintain and easy to make a rapid and timely progress in studies, which seems much more convenient, economic and efficient compared to mouse model. What is the limitation by using Drosophila model on studies, especially in metabolic diseases and cancer? Please at least comment or create a table to elaborate the differences.

Author Response

Comment 1 We thank the reviewer for drawing our attention to the Cell Metabolism article on A-FABP. We have included this new data and associated references in our manuscript.

Recent studies have begun to shed light on the potential molecular mechanisms that may link obesity with aggressive cancer progression and poor outcomes in breast cancer patients [54].  Emerging evidence suggests elevated levels of circulating adipose fatty acid binding protein (A-FABP) in obese patients correlates with elevated alcohol dehydrogenase 1 (ALDH1) levels in breast tumor cells and also with poor prognosis in obese breast cancer patients [54-56]. Injection of in vitro cultured breast tumor cells under the mammary fat pad of wildtype mice resulted in the release of A-FABP from adipose tissues and a concomitant increase in the expression of ALDH1 in transplanted tumor cells. Importantly, in GEMM that lacked A-FABP, ALDH1 expression levels and tumor growth were reduced in transplanted breast tumor cells. Furthermore, A-FABP-/- mutant mice bearing orthotopically injected breast cancer cells fed a high fat diet show reduced tumor burden compared to A-FABP-/- mice on a low fat diet. Finally, in a GEMM of post-menopausal breast cancer (MMTV-Tranforming Growth Factor) loss of A-FABP also showed reduced ALDH1 positive cells and decreased tumor growth [54]. These results suggest that circulating A-FABP enhances the aggressiveness of mammary tumor cells in both transplantation and transgenic mouse models. A-FABP activates the Interleukin 6/Signal Transducer and Activator of Transcription 3/ALDH1 pathway within breast tumor cells which promotes a stem cell phenotype that enhances the aggressiveness of these tumor cells [54].  Thus, A-FABP may represent a new link between obesity and increased risk of breast cancer and provide new diagnostic or therapeutic opportunities for patients.

Comment 2 We have incorporated a discussion paragraph and references to address Comment 2

2. As the authors’ claim that the Drosophila model are inexpensive, simple to maintain and easy to make a rapid and timely progress in studies, which seems much more convenient, economic and efficient compared to mouse model. What is the limitation by using Drosophila model on studies, especially in metabolic diseases and cancer? Please at least comment or create a table to elaborate the differences.

Notwithstanding the advantages of using Drosophila to investigate human cancers and metabolic disorders, there are also several limitations that should be considered. First, a clear physiological difference between flies and mammals is the optimal body temperature. Flies are ectotherms and can be successfully reared between 18 to 27oC whilst humans are endothermic and closely maintain a body temperature of approximately 37oC. As ectoderms do not have to maintain body temperature it has been proposed that these animals expend more energy on growth and reproduction than endoderms [61]. Hence, the relationship between diet, energy expenditure and animal growth between flies and humans may be biased towards different physiological outcomes. Second, flies lack a clear equivalent of breast, prostrate and lung tissues, thus preventing directly comparable models of these cancers from being generated. Nevertheless, human epithelial and stem cell cancers have been successfully modeled in Drosophila [26]. Drosophila also lack a closed circulatory system rendering the modeling of tumor induced angiogenesis and associated changes in the tumor microenvironment difficult. Finally, Drosophila models of drug discovery may be limited due to differences in drug metabolism pathways between flies and humans [62] [60].

Reviewer 2 Report

The manuscript named “Using animal models to investigate the mechanistic links between obesity, type II diabetes and cancer” reviews studies in mice cancer models in which either dietary or genetic manipulation has been used to model obesity or diabetes. As a completion the authors offer Drosophila as an unparalleled opportunity to combine dietary manipulation with models of human metabolic disease and cancer. The review is well-written and up-to-date. However, the authors do not describe another animal models, such as rats, widely used in cancer, diabetes or obesity. Thus, I would recommend the authors to re-name the manuscript to specify the only two animal models described in the manuscript.

In Introduction section, page 2, row 22 there is an inactive citation Zhang:2015fr. Please add the reference into the References.

In the section 2.2. Investigating interactions between diet and cancer in mice, there is an information about PubMed search. I putted the search words into PubMed, but what I found was different from the authors. For words “mouse cancer models and diet” I found 1621 items, for the words “mouse cancer models” 68704. The number of studies rises up fast, I would recommend the authors not to use such information because it is confusing.

In summary, the manuscript is very interesting and could have an impact on scientific community. After minor revisions I would recommend this manuscript to be accepted to the International Journal of Molecular Sciences.

Author Response

However, the authors do not describe another animal models, such as rats, widely used in cancer, diabetes or obesity. Thus, I would recommend the authors to re-name the manuscript to specify the only two animal models described in the manuscript.

Manuscript title now reads 'Using Mouse and Drosophila Models to Investigate the Mechanistic Links between Obesity, Type II Diabetes and Cancer.

      In Introduction section, page 2, row 22 there is an inactive citation Zhang:2015fr. Please add the reference into the References - corrected

      In the section 2.2. Investigating interactions between diet and cancer in mice, there is an information about PubMed search. I putted the search words into PubMed, but what I found was different from the authors. For words “mouse cancer models and diet” I found 1621 items, for the words “mouse cancer models” 68704. The number of studies rises up fast, I would recommend the authors not to use such information because it is confusing

Reference to Pubmed search deleted and text now reads.

      Despite the substantial human epidemiological evidence that reveals a clear association between cancer and diet, this area of human health is noticeably understudied. And yet, even with such limited study, increased tumor growth and more aggressive metastatic disease has been observed in animals reared on high calorie diets for a variety of mouse cancer models, including pancreatic [35], prostate [36,37], ovarian [38] and breast cancers [39]

Round 2

Reviewer 1 Report

The authors have addressed all my questions. I agree to accept this manuscript.